# Corneal Endothelial Cell Loss Following Cataract Surgery in Patients with Type 2 Diabetes Mellitus: A Comprehensive Review

**DOI:** 10.3390/biomedicines13071726

**Published:** 2025-07-15

**Authors:** Mădălina-Casiana Salavat, Mihnea Munteanu, Vlad Chercotă, Adina Iuliana Ardelean, Amanda Schuldez, Valentin Dinu, Ovidiu Borugă

**Affiliations:** 1Department IX Surgery, Discipline of Ophthalmology, Victor Babes University of Medicine and Pharmacy, E. Murgu Sq., No. 2, 300041 Timisoara, Romania; palfi.madalina@umft.ro (M.-C.S.);; 2Department VII Neurosciences, Neurology II Clinic, Victor Babes University of Medicine and Pharmacy, E. Murgu Sq., No. 2, 300041 Timisoara, Romania; 3Department XII, Discipline of Ophthalmology, “Carol Davila” University of Medicine and Pharmacy, 050474 Bucharest, Romania; 4Bucharest Emergency Eye Hospital, 030167 Bucharest, Romania

**Keywords:** cataract surgery, corneal endothelial cell loss, diabetes mellitus type 2, phacoemulsification, risk factors, surgical techniques, long-term consequences

## Abstract

Cataract surgery, while commonly considered a routine, highly effective, and generally low-risk ophthalmic procedure, has been associated with corneal endothelial cell loss (ECL), a phenomenon particularly pronounced in patients with type 2 diabetes mellitus (DM2). This increased susceptibility in diabetic patients is often attributed to pre-existing corneal abnormalities, including compromised structural integrity and reduced endothelial cell density. Additionally, metabolic stress factors inherent to diabetes, such as chronic hyperglycemia and associated oxidative stress, further exacerbate endothelial vulnerability. Consequently, diabetic patients may experience significantly greater endothelial cell loss during and after cataract surgery, necessitating targeted surgical strategies and careful perioperative management to preserve corneal health and visual outcomes. This paper aims to conduct an extensive and detailed review of the existing scientific literature to thoroughly investigate the relationship between ECL and cataract surgery in patients diagnosed with DM2. This study conducts a critical evaluation to elucidate the mechanisms contributing to high endothelial vulnerability in individuals with diabetes. It systematically compares the rates of ECL observed in diabetic and non-diabetic populations undergoing cataract surgery, examines molecular alterations following the procedure in patients with and without DM2, identifies key risk factors influencing surgical outcomes, evaluates the impact of various surgical techniques, discusses preventative measures, and examines the long-term consequences of ECL in this specific population. Furthermore, this review analyzes the existing research to identify gaps in knowledge and suggest potential directions for future investigations.

## 1. Introduction

Cataract, defined by the opacification of the crystalline lens, represents a primary global cause of visual impairment and blindness, significantly impacting public health and ocular medicine [1]. Cataract surgery, predominantly utilizing phacoemulsification and intraocular lens (IOL) implantation, serves as a highly effective intervention for the restoration of vision in individuals affected by lens opacification [2,3,4]. Individuals diagnosed with DM2 pose distinct considerations in the context of cataract surgery [5]. As a chronic metabolic disorder marked by persistent hyperglycemia, DM2 is associated with a range of microvascular complications, including ocular manifestations that may influence surgical outcomes and postoperative recovery [5]. Corneal abnormalities are frequently observed in individuals with diabetes, manifesting as heightened epithelial fragility, diminished corneal sensitivity, compromised barrier integrity, delayed wound healing, and an increased susceptibility to corneal edema [5]. These pre-existing ocular conditions may increase the vulnerability of the corneal endothelium in diabetic patients, increasing the risk of endothelial damage both intraoperatively and postoperatively in the context of cataract surgery [4,5,6,7].

This study aims to integrate and critically analyze the existing scientific literature on the relationship between ECL and cataract surgery in individuals with DM2. Through a comprehensive analysis of the existing literature, this review seeks to elucidate the underlying mechanisms and molecular alterations, compare surgical outcomes between diabetic and non-diabetic patients, identify key risk factors, assess the impact of surgical parameters, explore preventative measures, and examine the potential long-term consequences of increased ECL in this clinically vulnerable population. Also, this review aims to identify gaps in the existing body of knowledge and propose directions for future research, with the objective of enhancing surgical outcomes and reducing corneal decompensation.

## 2. Materials and Methods

This review employed a rigorous methodology, commencing with a systematic literature search and a comprehensive review of the selected literature. This integrated approach was designed to enhance the robustness of our findings and facilitate an in-depth exploration of the research question. The initial phase of our methodology focused on the systematic literature search strategy. We conducted comprehensive searches across reputable scientific databases, Scopus, Web of Science, and PubMed, ensuring the credibility and high quality of our source material. The search utilized a precise combination of keywords and Boolean operators, specifically the following: (“Corneal endothelium” OR “endothelial cell” OR “corneal endothelial cell count”) AND (“cataract surgery” OR “phacoemulsification” OR “intraocular lens implantation”) AND (“diabetes mellitus type 2” OR “type 2 diabetes” OR “T2DM”). This strategy aimed to identify original research articles published between 2001 and 2025, with the search terms targeted within the “title,” “keywords,” and “abstract” fields. To ensure relevance and accessibility, the search was limited to English-language publications, with a preference for open-access articles to maximize knowledge dissemination. In our pursuit of high-quality, peer-reviewed evidence, we excluded non-original works such as conference proceedings, theses, and encyclopedias, focusing exclusively on journal articles that directly contribute to the field.

To ensure the focus and rigor of this review, the following exclusion criteria were applied:Irrelevant outcomes: Studies that did not report on outcomes directly pertaining to corneal endothelial cell loss or density following cataract surgery were omitted.Type 1 diabetes mellitus: Studies focusing exclusively on patients with type 1 diabetes mellitus were excluded to maintain a clear focus on the implications for type 2 diabetes mellitus.Insufficient data: Articles lacking adequate data for synthesis or critical appraisal, such as those with incomplete methodologies or results, were excluded.

An overview of the study selection procedure is presented in Figure 1.

## 3. Background

### 3.1. Anatomy and Function of Corneal Endothelial Cells

The corneal endothelium consists of a monolayer of hexagonal cells situated on the posterior surface of the cornea, delineating the boundary with the anterior chamber of the eye [8]. This specialized cellular structure plays a crucial role in maintaining corneal transparency and regulating intraocular fluid dynamics [9]. Also, the cornea operates as an active pump, facilitating the transport of fluid from the corneal stroma in order to maintain a relatively dehydrated state essential for optical clarity [10]. With advancing age, there is a natural and progressive reduction in corneal endothelial cell density (ECD). At birth, the approximate central corneal endothelial cell density is 4000 cells/mm^2^. This density undergoes a more pronounced reduction within the initial two years of life, reaching approximately 3500 cells/mm^2^ by five years of age. In healthy adults aged 20–39 years, the average endothelial cell density is approximately 3000 cells/mm^2^. This decline continues with advancing age, exhibiting an average annual reduction rate of 0.3–0.6%. By 60–79 years of age, healthy individuals typically present an average endothelial cell density of 2600 cells/mm^2^. A critical functional threshold exists for corneal endothelial cell density. When this density falls below approximately 500 cells/mm^2^, the essential pump and barrier functions of the endothelium become compromised, leading to corneal endothelial dysfunction and a resultant decrease in corneal clarity [10]. A minimum threshold of approximately 400–500 cells/mm^2^ is essential to preserve adequate corneal hydration and transparency. This cellular reduction necessitates careful monitoring to ensure the maintenance of corneal function and ocular health [8]. In contrast to other corneal layers, endothelial cells in adults exhibit a restricted regenerative capacity. Consequently, cellular attrition is predominantly offset by compensatory mechanisms, including an increase in cell size (polymegathism) and modifications in cellular morphology (pleomorphism), ensuring the maintenance of corneal function [2,10]. A substantial reduction in endothelial cell density may result in corneal decompensation, manifesting as stromal edema, diminished transparency, and compromised visual acuity. This pathological process underscores the critical role of endothelial integrity in maintaining corneal homeostasis and optimal optical function [9].

### 3.2. Cataract Surgery Techniques

Phacoemulsification is the predominant surgical method for cataract extraction, characterized by a precise sequence of operative steps. The procedure entails the creation of a small corneal incision, followed by the formation of a capsulorhexis to facilitate access to the lens [2]. Subsequently, an ultrasonic handpiece is employed to fragment and emulsify the cataractous lens, enabling its efficient removal while preserving structural integrity within the anterior chamber [3].

Femtosecond laser-assisted cataract surgery (FLACS) represents a contemporary advancement in cataract extraction, integrating femtosecond laser technology to facilitate key surgical steps, including corneal incisions, capsulorhexis formation, and lens fragmentation [3]. Innovators of this technique propose that FLACS may reduce the necessity for excessive ultrasound energy during phacoemulsification, potentially minimizing endothelial cell damage and preserving corneal integrity [11].

### 3.3. Prevalence and Pathophysiology of Diabetes Mellitus Type 2

Diabetes mellitus represents an escalating global health challenge, with DM2 being the most prevalent form [1]. DM2 is distinguished by insulin resistance and relative insulin deficiency, culminating in persistent hyperglycemia [8]. This metabolic disorder exerts widespread systemic effects, notably compromising vascular and neural integrity, thereby predisposing individuals to complications such as diabetic retinopathy, nephropathy, neuropathy, and cardiovascular disease [7].

Within the ocular environment, persistent hyperglycemia facilitates the accumulation of advanced glycation end products (AGEs) across multiple tissues, including the cornea [2,9]. The presence of AGEs disrupts cellular function, exacerbates oxidative stress, and contributes to structural alterations within the corneal endothelium, potentially compromising its physiological integrity and function [9,11]. Moreover, chronic hyperglycemia can impair the activity of the sodium–potassium ATPase pump in corneal endothelial cells, a mechanism crucial for sustaining proper corneal hydration. These diabetes-induced disruptions may adversely affect endothelial cell function and structural integrity, thereby heightening susceptibility to surgical trauma during cataract extraction procedures [4].

## 4. Literature Review: Endothelial Cell Loss Following Cataract Surgery in Patients with Type 2 Diabetes Mellitus—A Synthesis of Current Findings

Extensive research has examined the effects of cataract surgery on corneal endothelial cells in individuals with type DM2. Numerous studies have documented a statistically significant reduction in ECD following phacoemulsification in diabetic patients, highlighting the potential implications for corneal health and postoperative recovery [3].

For example, one study reported a mean endothelial cell loss of 472.7 ± 369.1 cells/mm^2^ in the diabetic cohort, whereas the non-diabetic group exhibited a lower mean reduction of 165.7 ± 214.6 cells/mm^2^ within the first postoperative week. These findings underscore the differential impact of cataract surgery on corneal endothelial integrity in patients with diabetes mellitus [12].

Another study documented a mean reduction of 154 cells/mm^2^ (6.2%) in the diabetic cohort, compared to a decrease of 42 cells/mm^2^ (1.4%) in the control group at three months postoperatively. These findings highlight the differential impact of cataract surgery on corneal endothelial integrity between diabetic and non-diabetic patients [13].

These recommendations indicate that cataract surgery may lead to a more pronounced degree of endothelial cell loss in individuals with type DM2 compared to non-diabetic patients. However, some studies have reported contradictory results, demonstrating no statistically significant difference in ECL between diabetic and non-diabetic cohorts [2].

Pandey et al. observed higher endothelial cell loss in diabetic patients after cataract surgery, even in presence of good glycemic control, in comparison with non-diabetic patients [14].

For instance, one study observed substantial postoperative ECL in both diabetic and non-diabetic groups at one month following surgery; however, no significant differences were detected between the groups at any subsequent time point up to six months [2]. These inconsistencies may be attributable to variations in surgical techniques, patient demographics, disease severity, glycemic control, and the timing of endothelial cell assessment postoperatively [15,16]. A brief of all these changes can be seen in Table 1.

Systematic reviews and meta-analyses have sought to synthesize the existing body of evidence on this topic. One such review underscored the variability in reported visual outcomes among diabetic patients undergoing cataract surgery, reflecting the influence of diverse methodological approaches and patient-specific factors [1]. A study investigating FLACS observed that certain research findings indicated a lower degree of endothelial cell loss in healthy eyes when compared to conventional surgical techniques. These results suggest potential advantages of FLACS in preserving corneal endothelial integrity [11]. The existing body of literature indicates a tendency toward heightened ECL in diabetic patients following cataract surgery. However, the extent and reliability of this observed disparity necessitate further comprehensive investigation to establish definitive conclusions.

## 5. Comparative Analysis of Molecular Changes Following Cataract Surgery in Patients with and Without DM2

### 5.1. Molecular Changes in Non-Diabetic Patients After Cataract Surgery

Cataract surgery, even when uneventful, induces an acute inflammatory response in the eye due to surgical trauma. The mechanical manipulation of ocular tissues and the energy used in phacoemulsification lead to the release of pro-inflammatory mediators, including adhesion molecules, chemokines, cytokines, and growth factors, which facilitate the initial healing process. Notably, cytokines such as IL-6, IL-8, TNF-α, and MCP-1 reach peak levels within the first 24 h post-surgery in individuals without diabetes [17,18,19,20].

A noteworthy observation is the significant elevation of interleukin-6 (IL-6) concentrations within the aqueous humor following surgical intervention. In non-diabetic patients, this postoperative inflammatory response constitutes a well-characterized physiological reaction designed to initiate tissue repair and wound healing. The surgical trauma induces cellular damage, leading to the release of damage-associated molecular patterns (DAMPs). Subsequently, these DAMPs activate immune cells, thereby stimulating the production and secretion of pro-inflammatory cytokines and chemokines. These signaling molecules function to recruit further immune cells to the surgical site, facilitate the removal of cellular debris, and instigate the complex cascade of events essential for tissue regeneration [21,22].

### 5.2. Molecular Changes in Patients with Type 2 Diabetes Mellitus After Cataract Surgery

In individuals with type 2 diabetes mellitus, the postoperative inflammatory response subsequent to cataract surgery demonstrates significant variations compared to non-diabetic subjects, frequently manifesting as a reaction of heightened intensity and protracted duration. Research has suggested that diabetic patients may exhibit elevated baseline concentrations of specific inflammatory cytokines within the aqueous humor even prior to surgical intervention. Moreover, a marked increase in cytokines, including interleukin-1β (IL-1β), interleukin-6 (IL-6), interleukin-8 (IL-8), interferon-induced protein-10 (IP-10), monocyte chemoattractant protein-1 (MCP-1), and vascular endothelial growth factor (VEGF), has been observed in diabetic patients who subsequently develop macular edema following cataract surgery [20,21]. Analysis of the molecular profiles associated with wound healing reveals significant disparities between the two patient groups. Type 2 diabetes mellitus is known to impair corneal wound healing at a molecular level, involving alterations in the basement membrane, reduced corneal nerve fiber density, and compromised growth factor signaling [23]. Specifically, studies have observed the suppression of Wnt-5a, a crucial signaling protein in corneal epithelial wound healing, in diabetic corneas. This suppression is linked to DNA hypermethylation of the Wnt5A gene promoter and upregulation of miR-203a. Furthermore, in advanced diabetes, corneal stem cells exhibit dysfunction, contributing to a slower and less complete healing process following injury or surgical procedures such as cataract surgery [23,24]. Diabetic corneas also present with basement membrane thickening, multilamination, and reduced anchoring fibrils, which impair epithelial adhesion and contribute to abnormal wound healing. Additionally, in vivo confocal microscopy has revealed reduced sub-basal nerve density and increased nerve tortuosity in diabetic corneas, factors associated with delayed wound healing [25]. While specific data on wound healing molecular profiles in other ocular tissues after cataract surgery in both groups is limited within the provided snippets, the established impairment in diabetic patients suggests that the overall healing process following surgery might be less efficient and potentially delayed compared to non-diabetic individuals [22,23,25]. A schematic illustration of the molecular pathways influenced by cataract surgery in diabetic individuals is provided in Figure 2 [23,24,25].

## 6. Mechanisms Exacerbating Endothelial Cell Loss in Diabetic Patients Undergoing Cataract Surgery

Various physiological mechanisms have been postulated to elucidate the heightened ECL observed in patients with DM2 following cataract surgery. These proposed mechanisms aim to clarify the underlying factors contributing to this phenomenon, potentially facilitating improved surgical outcomes and patient management in ophthalmic care.

### 6.1. Pre-Existing Corneal Endotheliopathy

Persistent hyperglycemia associated with DM2 has been implicated in the development of structural and functional impairments within the corneal endothelium. These pathological alterations may contribute to compromised corneal integrity and overall ocular health, necessitating further investigation into their underlying mechanisms and potential therapeutic interventions [26]. The progressive accumulation AGEs has been associated with impairments in cellular adhesion and heightened oxidative stress, thereby compromising the structural integrity of the corneal endothelium. These pathological alterations render the endothelial cells more susceptible to surgical trauma, potentially influencing postoperative outcomes and necessitating further investigative and therapeutic considerations [2]. Patients with diabetes frequently present with reduced preoperative ECD and exhibit morphological alterations in endothelial cells, including pleomorphism and polymegethism. These deviations serve as indicators of a diminished endothelial reserve, potentially impacting corneal homeostasis and surgical outcomes [5].

### 6.2. Hyperglycemia and Metabolic Stress

In diabetic patients, elevated glucose concentrations within the aqueous humor have been shown to directly compromise endothelial cell function. Hyperglycemia can disrupt the activity of the sodium–potassium ATPase pump, a critical mechanism for preserving corneal deturgescence. This impairment may contribute to an increased risk of corneal edema in the postoperative period, potentially affecting visual recovery and overall surgical outcomes [4,27]. Moreover, the metabolic stress triggered by hyperglycemia can lead to an upregulation in the generation of reactive oxygen species (ROS), thereby exacerbating oxidative damage to endothelial cells both intraoperatively and postoperatively. This process may compromise corneal integrity and contribute to adverse surgical outcomes, highlighting the need for further investigation into potential protective strategies [8,28,29].

### 6.3. Increased Intraoperative Inflammation

Individuals with diabetes exhibit an increased susceptibility to intraocular inflammation following cataract surgery [30,31]. This heightened inflammatory response may be attributed to metabolic dysregulation, oxidative stress, and impaired immune function, all of which contribute to postoperative complications and may necessitate targeted therapeutic interventions [32]. The exacerbated inflammatory response observed in diabetic patients can further compromise the integrity of the already vulnerable corneal endothelium, leading to an increased rate of endothelial cell loss [26,33]. This process may contribute to postoperative complications, emphasizing the need for targeted therapeutic strategies to modify inflammation-induced endothelial damage [26,27,34].

### 6.4. Surgical Factors

Although not limited to individuals with diabetes, specific surgical factors may have a disproportionate impact on patients with pre-existing endothelial compromise. Among these factors, reduced pupil size—frequently observed in diabetic patients due to autonomic neuropathy—can exacerbate surgical complexity, increasing the likelihood of instrument–cornea contact and consequently heightening the risk of endothelial damage [9]. Furthermore, the ultrasound energy applied during phacoemulsification, which is directly associated with ECL, may exert a more significant impact on the structurally compromised endothelium of diabetic patients. This raised susceptibility underscores the importance of optimizing surgical parameters to reduce endothelial damage and preserve corneal integrity [13].

## 7. Risk Factors for Increased Endothelial Cell Loss in Diabetic Patients

Several risk factors have been identified that may contribute to increased ECL after cataract surgery in patients with DM2:

### 7.1. Severity and Duration of Diabetes

Several studies indicate that the duration of diabetes may serve as a critical determinant in the extent of ECL. Prolonged exposure to hyperglycemia and its associated metabolic disturbances may exacerbate endothelial dysfunction, thereby contributing to increased susceptibility to surgical trauma and postoperative complications. Further research is necessary to elucidate the precise mechanisms underlying this correlation and to develop targeted interventions [12]. Individuals with a prolonged history of diabetes are more likely to exhibit baseline endothelial compromise, rendering them increasingly vulnerable to surgical-induced damage. The cumulative effects of chronic hyperglycemia and associated metabolic dysregulation may exacerbate endothelial dysfunction, necessitating careful preoperative assessment and custom surgical approaches to diminish potential complications [24]. Conversely, some studies have reported no statistically significant association between the duration of diabetes and the extent of ECL. These findings suggest that additional factors may contribute to endothelial vulnerability, warranting further investigation to clarify the underlying mechanisms influencing postoperative outcomes in diabetic patients [5].

### 7.2. Glycemic Control

Suboptimal glycemic regulation, as reflected by elevated HbA1c levels, has been correlated with a greater extent of ECL in certain studies [27,28]. This association underscores the potential impact of chronic hyperglycemia on corneal endothelial integrity, necessitating further investigation into its pathophysiological mechanisms and clinical implications [5]. A study reported that the mean endothelial cell loss was markedly greater in patients with HbA1c levels exceeding 7% in comparison to those with HbA1c levels below 7%. These findings suggest a potential association between poor glycemic control and increased susceptibility to endothelial damage, warranting further investigation into the underlying mechanisms and clinical implications [5]. These findings indicate that preoperative optimization of glycemic control may play a critical role in reducing endothelial damage. Nevertheless, other studies have reported significant ECL in diabetic patients despite adequate glycemic regulation, suggesting that additional factors may contribute to endothelial vulnerability and postoperative outcomes [5,13].

### 7.3. Age

Advanced age is a well-documented risk factor for ECL following cataract surgery in both diabetic and non-diabetic populations. Age-related endothelial decline, coupled with potential systemic comorbidities, may contribute to increased susceptibility to surgical-induced endothelial damage, underscoring the need for perioperative management strategies [2]. The age-related reduction in ECD contributes to increased susceptibility of the corneal endothelium to surgical stress. This progressive decline in cellular integrity may exacerbate postoperative endothelial cell loss, highlighting the importance of personalized surgical approaches in older patients [4]. A study demonstrated that individuals aged 60–69 years exhibited a 3.8-fold higher risk of ECL compared to those in the 50–59-year age group within the control cohort. These findings underscore the impact of aging on endothelial vulnerability, emphasizing the need for customized surgery to address age-related endothelial damage [26].

### 7.4. Cataract Grade/Lens Density

Advanced cataract grades and increased lens nucleus density frequently necessitate higher levels of phacoemulsification energy, which may contribute to a greater extent of ECL. The intensified ultrasonic energy required for lens fragmentation can impose additional stress on the corneal endothelium, emphasizing the importance of optimizing surgical parameters to minimize endothelial damage and preserve corneal integrity [2]. A study demonstrated a significant correlation between higher cataract grades and a greater reduction in ECD at the six-month postoperative follow-up. These findings suggest that increased lens opacity may exacerbate surgical-induced endothelial damage [2].

### 7.5. Intraoperative Factors

Extended effective phaco time (EPT) and elevated cumulative dissipated energy (CDE) during phacoemulsification have been recognized as significant risk factors contributing to increased ECL. The prolonged exposure to ultrasonic energy may exacerbate endothelial damage, increasing the need for optimized surgical parameters to minimize corneal stress and preserve endothelial integrity [4]. A study conducted on non-diabetic patients revealed that individuals with an EPT of ≥0.50 min exhibited an 8.8-fold higher risk of ECL compared to those with an EPT of <0.25 min. These findings highlight the impact of prolonged ultrasonic exposure on endothelial integrity [26].

### 7.6. Preoperative Corneal Endothelial Density

Patients with reduced preoperative ECD are at an elevated risk of experiencing corneal decompensation following cataract surgery, particularly in cases where additional endothelial cell loss ensues [29]. This increased susceptibility underscores the critical role of preoperative endothelial assessment in optimizing surgical outcomes and decreasing postoperative complications [4].

### 7.7. Postoperative Inflammation

Heightened postoperative inflammation has been recognized as a contributing factor to ECL in diabetic patients [30]. This inflammatory response may exacerbate endothelial vulnerability, potentially influencing surgical outcomes and necessitating targeted anti-inflammatory interventions [30,31,32]. A study reported that diabetic patients presenting with an inflammatory score of 1+ during the first postoperative week exhibited a 5.7-fold higher risk of ECL compared to those with a score of 0.5+ [26].

## 8. Impact and Optimization of Surgical Strategies

### 8.1. Surgical Techniques

Phacoemulsification vs. Manual Small Incision Cataract Surgery (SICS): Several studies have examined ECL following phacoemulsification and SICS in patients with diabetes. One study reported a lower degree of endothelial cell loss associated with SICS compared to phacoemulsification in both diabetic and non-diabetic cohorts; however, the observed difference did not reach statistical significance [33]. Another study documented a progressive decline in ECD following SICS in diabetic patients, with a statistically significant reduction observed at the three-month follow-up [5]. The choice of surgical technique might influence the degree of ECL, but more comparative research is needed specifically in the diabetic population.

### 8.2. Conventional Phacoemulsification vs. Femtosecond Laser-Assisted Cataract Surgery (FLACS)

Multiple studies have explored the potential advantages of FLACS in preserving ECL relative to conventional phacoemulsification in patients with diabetes [33]. One study found that diabetic patients with moderate cataracts may exhibit a more pronounced reduction in endothelial cell density following conventional phacoemulsification compared to FLACS [12]. This phenomenon may be explained by the reduced phacoemulsification energy required in FLACS, as the laser pre-fragments the lens nucleus, facilitating a more efficient extraction process [33]. Another study, however, reported similar alterations in corneal endothelial cell density between diabetic and non-diabetic cohorts following FLACS, indicating that laser-assisted techniques may reduce the heightened risk typically associated with diabetes [32].

### 8.3. Advancements in Surgical Techniques

#### 8.3.1. Femtosecond Laser-Assisted Cataract Surgery (FLACS)

This automated technique frequently results in lower phacoemulsification energy requirements during the procedure, thereby potentially reducing mechanical trauma to the corneal endothelium when compared to conventional phacoemulsification [32]. In the study, specifically investigating diabetic patients, J.C.G. Cruiz et al. observed a statistically significant and clinically meaningful higher mean ECD of 423.55 cells/mm^2^ in the FLACS group compared to patients undergoing conventional phacoemulsification [33]. This suggests a potential advantage of FLACS in preserving endothelial cells in this vulnerable population. Furthermore, FLACS has been associated with improved visual rehabilitation and refractive outcomes [26]. The evidence supporting the superiority of FLACS in reducing endothelial cell loss in diabetic patients remains inconclusive, as multiple studies have reported comparable rates of ECL between FLACS and conventional phacoemulsification techniques [33]. A notable study suggested that FLACS may provide distinct advantages for patients with relatively dense cataracts and pre-existing low endothelial cell counts [34].

#### 8.3.2. Torsional Phacoemulsification

Employs a different mode of ultrasound energy delivery, utilizing a side-to-side oscillating motion of the phaco tip [34,35]. This technology may lead to decreased endothelial damage by enabling more efficient lens emulsification at lower energy levels and with reduced phacoemulsification time compared to traditional longitudinal ultrasound [36]. Some research suggests that torsional ultrasound is associated with lower endothelial cell loss and a faster recovery of visual acuity [35,36].

However, not all studies have corroborated these findings, with one study observing no significant difference in ECL between torsional and standard phacoemulsification in general cataract patients [33]. A study specifically examining diabetic patients found that, despite comparable intraoperative parameters between torsional and conventional phacoemulsification, the diabetic cohort undergoing torsional phacoemulsification demonstrated a significantly greater endothelial cell loss on both the first and thirtieth postoperative days [33].

#### 8.3.3. Eight-Chop Technique

The eight-chop technique is a phacoemulsification method in which the cataractous lens nucleus is manually divided into eight smaller segments prior to the application of ultrasound energy. This pre-segmentation strategy is designed to decrease dependence on phacoemulsification energy, thereby potentially reducing overall surgical trauma and endothelial cell loss [37,38,39]. A study examining the outcomes of this technique in diabetic patients found that endothelial cell density loss was minimal, with the rate of decline at one year postoperatively closely resembling that observed in non-diabetic patients [39]. The eight-chop technique has also been associated with shorter operative times and less invasive surgical manipulation [38,39].

#### 8.3.4. IOL-Shell Technique

It is a relatively novel surgical approach where the intraocular lens is utilized as a protective barrier for the corneal endothelium during phacoemulsification [39,40,41]. Research suggests that this technique results in less endothelial cell loss compared to conventional phacoemulsification [41]. However, its effectiveness and safety in the specific context of diabetic patients require further investigation.

### 8.4. Optimized Use of Ophthalmic Viscoelastic Devices (OVDs)

#### 8.4.1. Chondroitin Sulfate–Hyaluronic Acid (CS-HA) OVDs

OVDs serve a vital function in cataract surgery by preserving anterior chamber stability, safeguarding intraocular structures, and enhancing surgical precision [42]. CS-HA OVDs—including Viscoat, DuoVisc, and DisCoVisc—have exhibited promising benefits in protecting the corneal endothelium, particularly when compared to OVD formulations composed exclusively of hyaluronic acid (HA) or hydroxypropyl methylcellulose (HPMC). These advantages underscore the importance of selecting an appropriate OVD to optimize surgical outcomes and ensure superior ocular tissue preservation [42,43]. A meta-analysis of randomized controlled trials demonstrated a markedly reduced postoperative decline in endothelial cell density when CS-HA OVDs were utilized, in contrast to OVDs composed solely of HA or HPMC. These study highlights the potential benefits of CS-HA OVDs in preserving corneal endothelial integrity following cataract surgery [43]. The improved protective effect of CS-HA OVDs stems from their strong binding affinity to the corneal endothelium, which is enhanced by their triple negative charge, fostering molecular attraction and adhesion [42].

#### 8.4.2. Viscoadaptive OVDs

Viscoadaptive OVDs, characterized by their ability to display both cohesive and dispersive behaviors based on shear rate, may provide an advantage in reducing endothelial cell loss when compared to other OVD variants, including very low viscosity dispersives and super viscous cohesives [43,44].

#### 8.4.3. Soft-Shell Technique

The soft-shell technique employs a two-step approach, beginning with the application of a dispersive OVD—such as a CS-HA OVD—to form a protective barrier over the corneal endothelium. This is followed by the introduction of a cohesive OVD, which sustains anterior chamber depth and aids in surgical maneuvering. By capitalizing on the unique characteristics of each OVD type, this method optimizes endothelial protection, especially in cases involving compromised corneas [45].

#### 8.4.4. Intraoperative Medications

Antioxidants (e.g., Ascorbic Acid): The production of reactive oxygen species and free radicals during phacoemulsification plays a key role in endothelial cell damage. Investigating the use of antioxidants as intraoperative adjuncts has emerged as a potential approach to counteract this effect. Experimental studies in animal models have shown that incorporating ascorbic acid into the irrigation solution during phacoemulsification significantly decreases endothelial cell loss [46,47]. Moreover, clinical case reports indicate that the perioperative topical application of ascorbic acid may contribute to the prevention of corneal endothelial decompensation in individuals with pre-existing endothelial dysfunction [48,49].

#### 8.4.5. Postoperative Care Protocols

Topical Nonsteroidal Anti-inflammatory Drugs (NSAIDs): Postoperative inflammation is widely acknowledged as a contributing factor to endothelial cell loss, with diabetic patients being especially susceptible to its effects [47]. The postoperative administration of topical NSAIDs is advised to efficiently control inflammation and mitigate the risk of cystoid macular edema, a complication that tends to be more frequent and severe in patients with diabetes [47].

#### 8.4.6. Preoperative Optimization

Glycemic Control: Although the precise relationship between preoperative glycemic control and intraoperative endothelial cell loss is still being studied, maintaining optimal blood glucose levels before cataract surgery is regarded as a proactive approach to minimizing postoperative complications and potentially strengthening the corneal endothelium’s ability to withstand surgical stress [50,51,52]. Some studies indicate that poor glycemic control, as reflected by an HbA1c level greater than 7%, is associated with a higher degree of endothelial cell loss [6].

Preoperative Assessment: A thorough preoperative evaluation, incorporating specular microscopy to assess the corneal endothelium, is especially crucial for diabetic patients. This assessment enables the determination of baseline endothelial cell density and morphology, facilitating surgical planning and helping to identify individuals at elevated risk for postoperative corneal decompensation [5].

## 9. Supporting Evidence

The following Table 2 summarizes the statistical evidence supporting the efficacy of the proposed improvement suggestions in minimizing ECL during cataract surgery in patients with type 2 diabetes mellitus.

## 10. Discussion and Potential Area for Future Research

The statistical evidence presented in Table 2 underscores the potential benefits of several improvement strategies aimed at minimizing endothelial cell loss during cataract surgery in patients with type 2 diabetes mellitus. Notably, the adoption of FLACS has demonstrated a significant reduction in endothelial cell loss compared to conventional phacoemulsification in diabetic patients [33]. The documented mean increase of 423.55 cells/mm^2^ in endothelial cell density within the FLACS group underscores the potential of this technology in preserving the integrity of the diabetic endothelium [33]. This protective effect is likely attributable to the decreased phaco energy requirements and minimized surgical manipulation inherent to FLACS [34,35].

Diabetic patients are known to have more fragile corneal endothelial cells. Their corneas may be more susceptible to damage during surgery, and they can experience a greater reduction in endothelial cell density compared to non-diabetic individuals. This makes minimizing surgical trauma especially important for this patient population. The “eight-chop technique” is a refined phacoemulsification method distinguished by pre-nuclear segmentation. Unlike traditional phacoemulsification techniques where the cataract is broken up primarily during the ultrasound process, the eight-chop technique involves mechanically dividing the lens nucleus into eight smaller segments before significant ultrasound energy is applied [38]. By pre-chopping the nucleus into multiple, smaller fragments, the surgeon needs to use significantly less ultrasound energy to emulsify and aspirate the cataract. Ultrasound energy is a major source of stress and damage to corneal endothelial cells. Dividing the nucleus into eight smaller pieces allows for more efficient removal, potentially leading to shorter operative times. Shorter surgical times also contribute to reduced exposure of the eye to intraoperative stressors [38,53,54,55].

Selecting the most suitable OVDs is a critical factor in surgical outcomes [54]. Meta-analytical findings indicate that CS-HA OVDs offer superior corneal endothelial protection compared to HA-only or HPMC OVDs, leading to a significantly lower postoperative reduction in endothelial cell density [43]. The superior binding affinity of CS-HA OVDs to the corneal endothelium enhances stability, forming a more effective protective barrier throughout the surgical procedure [43,44].

Investigating the use of intraoperative pharmacological agents, particularly antioxidants, represents a promising strategy in reducing endothelial damage. Preclinical research indicates that ascorbic acid significantly reduces endothelial cell loss, likely due to its free radical scavenging capabilities [48]. While further clinical research is warranted, the use of antioxidants during surgery could offer an additional layer of protection, particularly in the oxidative stress-prone diabetic eye.

While our review synthesizes current evidence on ECL following cataract surgery in patients with DM2, it is imperative to critically appraise the limitations within the existing body of literature. A significant gap identified is the paucity of high-quality comparative studies, particularly randomized controlled trials, directly comparing the impact of FLACS versus conventional phacoemulsification specifically within the diabetic patient population. Most studies included in this review are observational, retrospective, or include mixed patient cohorts, making it challenging to isolate the specific effects of surgical technique in diabetic eyes, which are known to have compromised endothelial cell function. The limited number of studies, coupled with heterogeneous methodologies, varying follow-up periods, and diverse measures of ECL, prevents definitive conclusions regarding the superiority of one technique over another in this vulnerable patient group. A significant finding from our critical appraisal of the literature is the dearth of high-quality comparative studies, particularly well-designed randomized controlled trials or large prospective cohort studies, or directly addressing, e.g., the efficacy and safety of FLACS versus conventional phacoemulsification in diabetic patients. The existing evidence relies heavily on retrospective analyses or subgroup analyses from broader studies, which limits the generalizability and robustness of conclusions regarding this specific patient population. This gap is particularly concerning given the increasing prevalence of diabetes and its known impact on ocular health and surgical outcomes.

Also, additional studies are required to comprehensively clarify the specific mechanisms through which diabetes intensifies endothelial cell loss both intraoperatively and postoperatively in cataract surgery. Investigating the complex interactions among hyperglycemia, AGE accumulation, oxidative stress, inflammation, and genetic predisposition may facilitate the development of more precise preventive and therapeutic approaches.

## 11. Conclusions

The current body of evidence suggests that patients with DM2 may be at an increased risk of corneal endothelial cell loss following cataract surgery compared to non-diabetic individuals. This increased vulnerability is likely due to a combination of pre-existing corneal abnormalities, metabolic stress induced by hyperglycemia, and a potentially heightened inflammatory response to surgical trauma. While some studies have shown comparable ECL rates, a significant number indicates a greater loss in diabetic patients.

Several risk factors, including the severity and duration of diabetes, poor glycemic control, older age, higher cataract grade, and increased intraoperative energy usage, have been associated with greater ECL in this population. The choice of surgical technique, particularly the potential benefits of femtosecond laser-assisted surgery in reducing ultrasound energy, warrants further investigation in diabetic patients. Preventative measures, such as optimizing glycemic control, employing careful surgical techniques, utilizing appropriate viscoelastic substances, and potentially incorporating antioxidant agents, are crucial for minimizing endothelial damage.

Future research should focus on addressing the identified knowledge gaps through well-designed comparative studies, subgroup analyses, and investigations into the underlying mechanisms of diabetes-related endothelial dysfunction. The development of predictive models could also aid in identifying high-risk patients and tailoring surgical management accordingly. Ultimately, a better understanding of the correlation between ECL and cataract surgery in patients with DM2 will contribute to improved surgical outcomes and the long-term preservation of corneal health and vision in this growing patient population.

## Figures and Tables

**Figure 1 biomedicines-13-01726-f001:**
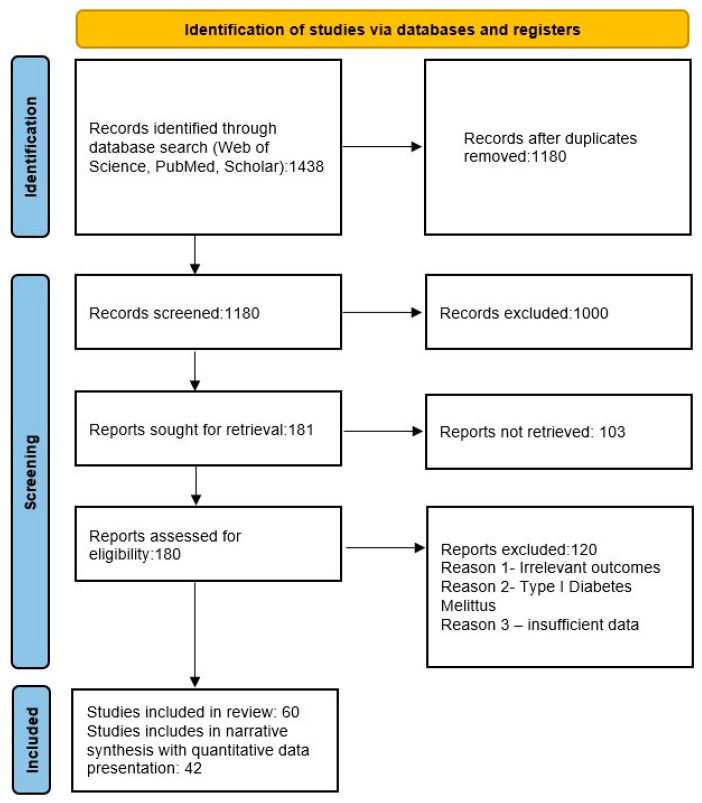
Flow chart of study selection process.

**Figure 2 biomedicines-13-01726-f002:**
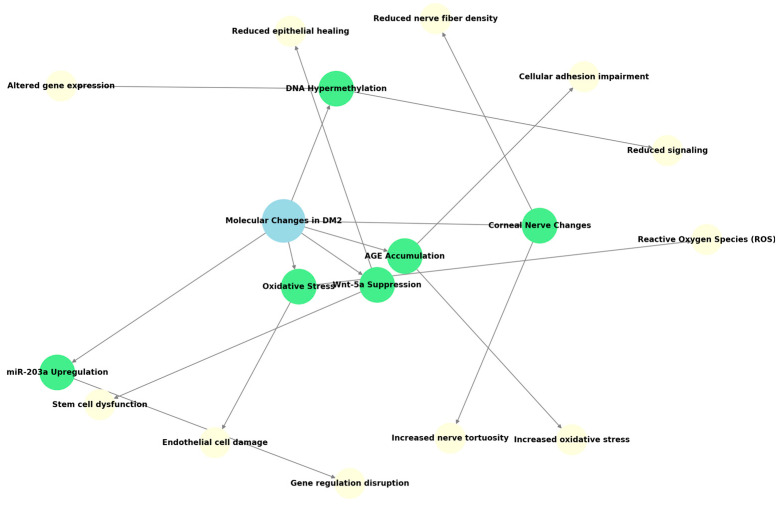
Molecular pathways impacted in diabetic patients post-cataract surgery (blue pathways: normal physiological mechanisms, green pathways: protective mechanisms, yellow pathways: stress-activated responses specific to diabetic corneas).

**Table 1 biomedicines-13-01726-t001:** Endothelial cell loss following cataract surgery in patients with DM2: a literature review.

Study Reference	Patient Group	Mean ECL (Cells/mm^2^)	Mean ECL (%)	Follow-Up Period	Significance (*p*-Value)	Patient Demographics (n, M/F)
Ciorba L.A. et al. [12]	Diabetic	472.7 ± 369.1	Not reported	1 week	<0.001	80 patients (40DM, 40 non-DM), sex not reported
Ciorba L.A. et al. [12]	Non-Diabetic	165.7 ± 214.6	Not reported	1 week	<0.001	
Hugod M. et al. [13]	Diabetic	154	6.2%	3 months	<0.05	124 patients (62 DM, 62 non-DM), sex not reported
Hugod M. et al. [13]	Non-Diabetic	42	1.4%	3 months	<0.05	
Pandey S. et al. [14]	Diabetic	Not reported	12.04%	12 weeks	<0.01	130 patients (65DM, 65 non-DM), sex not reported
Pandey S. et al. [14]	Non-Diabetic	Not reported	7.09%	12 weeks	<0.01	

**Table 2 biomedicines-13-01726-t002:** Strategies for minimizing ECL during cataract surgery.

Improvement Suggestion	Study	Patient Group	Mean ECL (Cells/mm^2^)	Mean ECL (%)	*p*- Value	% Reduction in ECL	Patient Demographics (nr, M/F)
FLACS vs. Conventional Phaco	Cruz et al. [34]	Diabetic	FLACS: 2118.17		<0.001	22.8%	95, sex not reported
			PHACO: 1628.479				
CS-HA OVDs vs. Other OVDs	Hsiao et al. [44]	General Cataract			<0.0001	4.10%	12 RCTs (n > 20 per group), no sex breakdown
						5.81%	
Good Glycemic Control (HbA1c < 7%) vs. Poor Control	Mazhar et al. [5]	Diabetic	Good: 187.79	8.63%	0.01	8.2%	355/42%M, 58%F
			Poor: 205.67	9.45%

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
