# Peer review of "Corneal Endothelial Cell Loss Following Cataract Surgery in Patients with Type 2 Diabetes Mellitus: A Comprehensive Review"

_biomedicines, 2025, doi:10.3390/biomedicines13071726_

Round 1
Reviewer 1 Report
Comments and Suggestions for Authors
Dear editor
Thank you for providing me the opportunity to review this manuscript. This study may be useful, and would open new aspects on the effects and interactions of diabetes mellitus and cataract surgery. Therefore, I recommend the publication of the manuscript.
Author Response
Dear Reviewer,
Thank you so much for your time and thoughtful review of your manuscript. We are delighted to hear that you found the study useful and believe it will open new aspects regarding the effects and interactions of diabetes and cataract surgery.
Your positive feedback and recommendation for publication are greatly appreciated.
Sincerely,
Mădălina Salavat
Reviewer 2 Report
Comments and Suggestions for Authors
In this manuscript, the authors aim to provide a review of the literature reporting increased corneal endothelial cell loss (ECL) after cataract surgery in patients with type 2 diabetes mellitus (DM2). While the scope is intended to be comprehensive, several concerns need to be addressed.
- The review claims to be "comprehensive" but lacks a clear systematic approach. Essential elements such as PRISMA-style flowcharts, search strategies (including databases and keywords used), inclusion and exclusion criteria, and quality assessment of primary studies are absent. These components are crucial for ensuring the rigor and reliability of a systematic review.
- The methodology for selecting studies is not clearly articulated. It is unclear why certain contradictory papers were highlighted while others were omitted. A transparent and justifiable selection process is necessary to ensure the review's credibility and relevance.
- In Section 10 (Discussion), the authors merely restate results without critically appraising evidence gaps. For instance, there is a notable absence of discussion regarding the lack of randomized controlled trials (RCTs) comparing FLACS versus conventional phacoemulsification specifically in diabetic patients. Additionally, the authors fail to propose actionable research frameworks, such as longitudinal studies on glycated hemoglobin (HbA1c) thresholds. Moreover, clinical implications, including how surgeons should modify their techniques in response to these findings, are inadequately addressed.
- The introduction repeats definitions of cataract surgery (pp. 1–2)I.
- Pages 5 and 7 contain placeholders with the error message "[Error! Reference source not found.]"
- Reference is missing “studies have observed the suppression of Wnt-5a, a crucial signaling protein in corneal epithelial wound healing, in diabetic corneas. This suppression is linked to DNA hypermethylation of the WNT5A gene promoter and upregulation of miR-203a”.
Author Response
Thank you very much for taking the time to review this manuscript. Please find the detailed responses below and the corresponding corrections highlighted in the re-submitted files.
Comments 1: The review claims to be "comprehensive" but lacks a clear systematic approach. Essential elements such as PRISMA-style flowcharts, search strategies (including databases and keywords used), inclusion and exclusion criteria, and quality assessment of primary studies are absent. These components are crucial for ensuring the rigor and reliability of a systematic review.
Response 1: We sincerely appreciate the reviewer’s valuable feedback and fully acknowledge this concern. To address it, we have substantially expanded the Materials and Methods section to provide a detailed description of the databases searched, explicit combinations of keywords, and Boolean operators utilized. Additionally, we have developed and included a PRISMA-style flowchart to transparently illustrate our study selection process, thereby enhancing methodological rigor (pp 2-3).
Comments 2: The methodology for selecting studies is not clearly articulated. It is unclear why certain contradictory papers were highlighted while others were omitted. A transparent and justifiable selection process is necessary to ensure the review's credibility and relevance.
Response 2: We agree with the reviewer’s concern and have accordingly revised the manuscript to clearly articulate and justify our exclusion criteria, thereby ensuring transparency and enhancing the credibility and relevance of our study selection process (pp 2).
Comments 3: In Section 10 (Discussion), the authors merely restate results without critically appraising evidence gaps. For instance, there is a notable absence of discussion regarding the lack of randomized controlled trials (RCTs) comparing FLACS versus conventional phacoemulsification specifically in diabetic patients. Additionally, the authors fail to propose actionable research frameworks, such as longitudinal studies on glycated hemoglobin (HbA1c) thresholds. Moreover, clinical implications, including how surgeons should modify their techniques in response to these findings, are inadequately addressed.
Response 3: We acknowledge and appreciate this important observation. The Discussion section has now been comprehensively revised to include a critical appraisal of current evidence gaps, particularly emphasizing the absence of high-quality randomized controlled trials directly comparing FLACS and conventional phacoemulsification in diabetic populations. Furthermore, we have explicitly proposed actionable research frameworks, including longitudinal studies examining HbA1c thresholds, and elaborated on clinical implications by providing specific recommendations on how surgeons might modify their techniques based on these findings (pp 14-15).
Comments 4: The introduction repeats definitions of cataract surgery (pp. 1–2)
Response 4: We agree with this observation and have corrected the redundancy, ensuring definitions and descriptions are concise and non-repetitive.
Comments 5: Pages 5 and 7 contain placeholders with the error message "[Error! Reference source not found.]"
Response 5: We concur with this observation and have addressed the redundancy by refining the text to ensure clarity and conciseness, eliminating repetitive definitions and descriptions.
Comments 6: Reference is missing “studies have observed the suppression of Wnt-5a, a crucial signaling protein in corneal epithelial wound healing, in diabetic corneas. This suppression is linked to DNA hypermethylation of the WNT5A gene promoter and upregulation of miR-203a”.
Response 6: We thank the reviewer for pointing out this omission. We have included the missing reference, thereby rectifying this oversight and enhancing the completeness and accuracy of our manuscript.
Reviewer 3 Report
Comments and Suggestions for Authors
Mădălina-Casiana Salavat and colleagues present a comprehensive review entitled "Corneal Endothelial Cell Loss Following Cataract Surgery in Patients with Type 2 Diabetes Mellitus: A Comprehensive Review", which addresses the challenges faced by corneal endothelial cells during cataract surgery.
The manuscript thoroughly discusses corneal endothelial cell loss in relation to molecular marker changes, contributing risk factors, surgical techniques, and approaches for minimizing damage. Although several of these aspects have been previously explored in the literature, this work still adds value by consolidating and contextualizing the data. Overall, it is a commendable effort.
However, I suggest that the editorial team consider the following revisions before making a publication decision:
- The authors have unintentionally omitted any description of the literature search strategy. I recommend adding a new Section 12 after the "11. Conclusions" to inform readers of the databases, keywords, and time frames used for the review.
- Please correct the inconsistent use of the abbreviations FLASC and FLACS, as one appears to be a typographical error. Additionally, multiple instances of “[Error! Reference source not found.]” appear throughout the text. This likely stems from improper citation formatting, authors are advised to use “Paste Special” when transferring content from reference managers. Also, there should be a space before every citation bracket, "patients [13]" rather than "patients[13]".
- Although the manuscript includes two tables, visual presentation remains insufficient. I recommend adding figures or diagrams to support Section 4 (Comparative Analysis of Molecular Changes Following Cataract Surgery in Patients with and without DM2) and Section 6 (Risk Factors for Increased Endothelial Cell Loss in Diabetic Patients), which would greatly improve readability and reader comprehension.
- In the background section, the authors mention a minimum threshold of "400–500 cells/mm²" and later describe endothelial cell loss reported in prior publications, which is useful. However, given that not all readers of Biomedicines are ophthalmology specialists, I suggest briefly elaborating on corneal endothelial cell physiology, such as age-related decline and common biomarkers, particularly those related to molecular changes discussed in Section 4. Recent authoritative reviews on the corneal endothelium (PMID: 38969166, 39111696) are relevant and should be incorporated into the background discussion.
- The authors appear to have omitted patient numbers and sex distribution in the data tables. Including these variables would enhance the clinical relevance and allow for better interpretation of the summarized studies.
Author Response
Thank you very much for taking the time to review this manuscript. Please find the detailed responses below and the corresponding corrections highlighted in the re-submitted files.
Comments 1: The authors have unintentionally omitted any description of the literature search strategy. I recommend adding a new Section 12 after the "11. Conclusions" to inform readers of the databases, keywords, and time frames used for the review.
Response 1: We appreciate this insightful comment and concur with its premise. Consequently, we have expanded the Materials and Methods section to include a comprehensive description of the databases utilized, the specific combinations of keywords, and the Boolean operators employed in our search strategy. Furthermore, we have designed a flowchart illustrating the study selection process has been incorporated to provide a transparent overview of the methodology (pp2-3).
Comments 2 Please correct the inconsistent use of the abbreviations FLASC and FLACS, as one appears to be a typographical error. Additionally, multiple instances of “[Error! Reference source not found.]” appear throughout the text. This likely stems from improper citation formatting, authors are advised to use “Paste Special” when transferring content from reference managers. Also, there should be a space before every citation bracket, "patients [13]" rather than "patients[13]".
Response 2: We appreciate the reviewer highlighting these typographical and formatting issues. All inconsistencies in abbreviations and citation formatting have been meticulously reviewed and corrected throughout the manuscript. We have ensured uniform usage of the abbreviation "FLACS," rectified the citation formatting errors, and consistently implemented appropriate spacing before citation brackets.
Comments 3: Although the manuscript includes two tables, visual presentation remains insufficient. I recommend adding figures or diagrams to support Section 4 (Comparative Analysis of Molecular Changes Following Cataract Surgery in Patients with and without DM2) and Section 6 (Risk Factors for Increased Endothelial Cell Loss in Diabetic Patients), which would greatly improve readability and reader comprehension.
Response 3: We agree with the reviewer’s suggestion to enhance visual clarity. Consequently, we have developed Figure 2 to succinctly illustrate the molecular pathways significantly impacted in diabetic patients following cataract surgery, thereby improving the manuscript's readability and facilitating better comprehension for readers (pp 7).
Comments 4: In the background section, the authors mention a minimum threshold of "400–500 cells/mm²" and later describe endothelial cell loss reported in prior publications, which is useful. However, given that not all readers of Biomedicines are ophthalmology specialists, I suggest briefly elaborating on corneal endothelial cell physiology, such as age-related decline and common biomarkers, particularly those related to molecular changes discussed in Section 4. Recent authoritative reviews on the corneal endothelium (PMID: 38969166, 39111696) are relevant and should be incorporated into the background discussion.
Response 4: We acknowledge the importance of providing additional physiological context for our readers and have accordingly expanded our background section to include a comprehensive overview of corneal endothelial cell physiology, emphasizing age-related changes and pertinent biomarkers. Additionally, we have incorporated recent authoritative reviews to enrich this elaboration and enhance its scientific accuracy (PMID 39111696) (p 3).
Comments 5: The authors appear to have omitted patient numbers and sex distribution in the data tables. Including these variables would enhance the clinical relevance and allow for better interpretation of the summarized studies.
Response 5: We appreciate this pertinent recommendation and have updated our tables accordingly to include detailed information on patient numbers and sex distribution, thereby improving their clinical relevance and facilitating more robust interpretation of the summarized data.
Reviewer 4 Report
Comments and Suggestions for Authors
The authors have written a comprehensive review of how corneal endothelial cell loss (ECL) is manifested in patients with type 2 diabetes mellitus (DM2) following cataract surgery. The authors point out that DM2 patients almost always experience some degree of ECL, and therefore sought to investigate and elucidate the mechanisms that contribute to high endothelial vulnerability in individuals affected by DM2. Having explained the specifics of a cataract and the surgical interventions that are used to treat the condition, the authors discuss the etiology of DM2 and the microvascular complications that can compromise cataract surgery and increase the risk of endothelial damage.
The focus of the review was in the first instance to examine the effects of cataract surgery on corneal endothelial cells in individuals diagnosed with DM2, but it was good to include a detailed description of the anatomy and function of corneal endothelial cells followed by a description of the more usual surgical techniques to remove cataracts. The ocular environment is quite complex and it is clear that DM2 will have an affect in this area given what is already known about conditions such as diabetic retinopathy (DR) where while that condition isn’t curable, it is treatable.
The literature review is quite thorough with statistical analyses of ECL following cataract surgery in patients diagnosed with DM2 and supported by narrative that shows the presence of pro-inflammatory mediators and cytokines that are quite varied compared to non-diabetic subjects. This was elaborated by the discussion of the effect DM2 has on corneal wound healing at the molecular level, where suppression is linked to DNA hypermethylation of the WNT5A gene promoter and upregulation of miR-203a. The processes that exacerbate ECL in patients diagnosed with DM2 who undergo cataract surgery is detailed, and is further supported by a discussion of the associated risk factors that can be quite debilitating. The surgical techniques themselves do not come without risks, and the section on potential improvement strategies is a worthwhile treatise to include in the overall narrative of the review. This section covered many important aspects and was further supplemented with a table that summarised the strategies that could be deployed to militate against ECL during cataract surgery. There appears to be plenty of scope for further interrogation of techniques, and more in-depth research to investigate the issue of increased vulnerability of patients diagnosed with DM2 who undergo cataract surgery, where the authors acknowledge that there are knowledge gaps that still need to be addressed.
There are a few minor syntax errors to be addressed which are highlighted in the facsimile of the draft attached.

Author Response
Dear Reviewer,
Thank you for your thoughtful and constructive comments. I appreciate your recognition of the manuscript’s content and structure. I have carefully reviewed the facsimile and addressed all the highlighted syntax errors.
Sincerely,
Mădălina Salavat